# Human umbilical cord mesenchymal stem cell-derived exosomes mitigate acute radiation-induced intestinal oxidative damage via the Nrf2/HO-1/NQO1 signaling pathway

Hongyu Wang[1,2], Jinbao Wang[1,2], Gaosheng Yang[1,2], Yanjie Li[1,2], Weikai Chen[2], Jianping Yu[2]*, Xiaopeng Han[2]*

1 The First School of Clinical Medicine, Gansu University of Chinese Medicine, Lanzhou, China,
2 Department of General Surgery I, Gansu Provincial Central Hospital, Lanzhou, China

* hanxiaopeng74@163.com (XH); yujianpingld@163.com (JY)

## Abstract

Acute radiation-induced intestinal injury (ARII), a prevalent complication of abdominal radiotherapy, remains clinically challenging due to limited therapeutic options. This study demonstrates the therapeutic efficacy of human umbilical cord mesenchymal stem cell-derived exosomes (hucMSC-Exos) in mitigating ARII through Nrf2/HO-1/NQO1 pathway activation. In a rat model receiving 12 Gy abdominal irradiation, systemic hucMSC-Exos administration significantly restored intestinal mucosal integrity and reduced oxidative damage markers. Mechanistically, hucMSC-Exos potentiated the antioxidant axis by upregulating Nrf2 signaling, as evidenced by histopathological, biochemical, and molecular analyses. Complementary in vitro experiments revealed hucMSC-Exos protected irradiated IEC-6 cells from oxidative dysfunction while enhancing proliferation, effects substantially attenuated upon Nrf2 silencing via siRNA. These findings establish that hucMSC-Exos orchestrate redox equilibrium through targeted Nrf2 pathway modulation, effectively counteracting radiation-induced enterocyte apoptosis. The elucidated mechanism expands the therapeutic paradigm of MSC-derived exosomes in radioprotection and provides a clinically translatable strategy for managing ARII in oncological radiotherapy.

## Introduction

Radiotherapy is one of the indispensable treatments for malignant tumors, alongside surgery and chemotherapy. However, its application inevitably causes damage to healthy tissues and cells [1]. The intestine, being highly sensitive to radiation, is particularly susceptible to damage during abdominal or pelvic radiotherapy, leading to radiation-induced intestinal injury (RII). Research indicates that acute radiation-induced intestinal injury (ARII) typically occurs within three months of

**Data availability statement:** All relevant data are within the paper and its Supporting Information files.

**Funding:** This work was supported by the Natural Science Foundation of Gansu Province (grant No. 23JRRA1386 to XH), the Lanzhou Science and Technology Bureau (No. 2023-4-65 to JY) and Gansu Provincial Youth Science and Technology Fund (grant No. 24JRRA1119to WC).

**Competing interests:** The authors have declared that no competing interests exist.

radiotherapy, with symptoms such as diarrhea, nausea, mild bleeding, and mal-absorption, affecting approximately 60–80% of patients [2,3]. In contrast, chronic radiation-induced intestinal injury (CRII) typically manifests between three months and several years post-therapy, with persistent symptoms, loss of gastrointestinal function, irreversible fibrosis, and even intestinal perforation leading to patient death [4,5]. Therefore, early intervention is crucial to improve the quality of life in ARII patients. However, the mechanisms underlying ARII remain incompletely understood, and there is no established optimal treatment, with many existing therapies showing limited efficacy.

Exosomes are extracellular vesicles (EVs) ranging from 40 to 160 nm in diameter, secreted and released by most cells [6,7]. As natural nanomaterials, exosomes can transfer specific proteins, lipids, RNA, and DNA between cells, with advantages such as low immunogenicity, high stability, ease of storage, and efficient delivery [8,9]. Human umbilical cord mesenchymal stem cell-derived exosomes (hucMSC-Exos) serve as key mediators of the biological effects of stem cell-secreted factors [10,11]. They not only retain the unique tissue repair, regenerative, and differentiation potentials of mesenchymal stem cells, but also overcome the limitations of allogeneic mesenchymal stem cells in clinical applications, such as potential immune rejection, tumor formation risks, ethical concerns, and challenges related to storage and transport [12,13]. Therefore, we propose the use of hucMSC-Exos as a candidate therapeutic for ARII.

Current studies indicate that ionizing radiation (IR) directly damages biomolecules and indirectly induces the generation of reactive oxygen species (ROS), disrupting cellular redox balance and causing oxidative damage to biomacromolecules such as DNA, lipids, and proteins [1,14,15]. Nuclear factor (erythroid-derived 2)-like 2 (Nrf2) is a nuclear transcription factor that mediates cellular defense mechanisms against oxidative stress-induced toxicity [16]. Kelch-like ECH-associated protein 1 (Keap1) regulates Nrf2 through ubiquitination and proteasomal degradation to maintain its inactive state. During oxidative stress, Keap1 is inactivated, leading to Nrf2 accumulation and nuclear translocation, which activates the expression of downstream antioxidant genes such as heme oxygenase-1 (HO-1) and NAD(P)H quinone dehydrogenase 1 (NQO1), a critical mechanism for cellular defense against oxidative damage [16–18]. Furthermore, studies have confirmed that stem cell-derived exosomes can serve as effective activators of Nrf2 [19]. Therefore, we hypothesize that hucMSC-Exos can counteract radiation-induced intestinal oxidative damage through the Nrf2/HO-1/NQO1 pathway and promote the restoration of the intestinal mucosal barrier.

This study validates the efficacy of hucMSC-Exos through in vitro and in vivo experiments and explores their potential mechanisms and pathways of action.

## Materials and methods

This study was conducted in compliance with the ARRIVE Guidelines 2.0 and the GB/T 35892−2018 guidelines for the ethical review of experimental animal welfare. All procedures were approved by the Laboratory Animal Ethics Committee of Gansu University of Chinese Medicine and performed in compliance with the approved guidelines (License No. SY2024−212).

## Reagents

All reagents used in this study were commercially available products. DMEM medium, phosphate-buffered saline (PBS), 10,000 U/mL penicillin-streptomycin, 0.25% trypsin-EDTA, and MesenPRO RS(TM) Basal Medium (Gibco, California, USA); exosome-depleted fetal bovine serum was purchased from System Biosciences (SBI, California, USA); primary antibodies against CD73, CD90, CD105, CD34, CD45, CD9, CD81, CD63, TSG101, and Calnexin (Abcam, Cambridge, UK); Cell Counting Kit-8 (CCK8), Hoechst 33342 stain, and DAB kit (Boster, Wuhan, China); superoxide dismutase (SOD), malondialdehyde (MDA), and glutathione peroxidase (GSH-Px) kits (Cayman Chemical, Michigan, USA); radioimmunoprecipitation assay (RIPA) buffer, bicinchoninic acid (BCA) protein assay kit, DAPI, reactive oxygen species (ROS) assay kit, crystal violet, and sodium dodecyl sulfate polyacrylamide gel electrophoresis (SDS-PAGE) (Beyotime, Shanghai, China); primary antibodies against Ki67, Nrf2, HO-1, and NQO1, and secondary antibodies for western blotting (Proteintech, Wuhan, China); small interfering RNA (siRNA) targeting Nrf2 (Tsingke, Beijing, China); RNAFit RNA-specific transfection reagent (Hanbio, Shanghai, China); M5 Universal RNA Mini Kit, M5 Sprint qPCR RT Kit with genomic DNA (gDNA) remover, and 2X M5 Hiper SYBR Premix Estaq (Mei5bio, Beijing, China). All other chemicals used in this study were of analytical grade.

## Cell culture and treatment

In this study, we used rat intestinal epithelial cells (IEC-6). The IEC-6 cell line was purchased from the Basic Medical Cell Center, Institute of Basic Medical Sciences, Chinese Academy of Medical Sciences. The cells were cultured in DMEM supplemented with 10% exosome-free fetal bovine serum (FBS) and 1% penicillin-streptomycin (10,000 U/mL) at 37°C in a 5% CO2 incubator, with media changes every three days. When cells reached 80–90% confluence, they were digested with 0.25% trypsin and transferred to new culture flasks for further expansion. The cells were cultured up to the third passage for experimentation.

## Extraction and identification of hucMSC-Exos

Human umbilical cord mesenchymal stem cells (hucMSCs) were purchased from Beijing Aomi Jiade Pharmaceutical Technology Co., Ltd. The cells were cultured in MesenPRO RS(TM) Basal Medium under conditions identical to those used for IEC-6 cells. Flow cytometric analysis was performed to identify hucMSC-specific markers (CD73, CD90, and CD105) and hematopoietic stem cell markers (CD34 and CD45) for confirmation [20]. When hucMSCs reached 80% confluence, they were incubated with 0.25% trypsin-EDTA solution for 2 minutes. After adding 2 mL of medium to stop trypsin digestion, the cells were centrifuged at 1000 rpm for 5 minutes. The cells were then cultured in exosome-free medium at a concentration of $1 \times 10^5$/mL, with medium changes every three days until confluence. The conditioned medium supernatant was collected and processed using a gradient ultracentrifugation method. Briefly, the supernatant was centrifuged at 500 g for 15 minutes to remove cells, at 16,500 g for 20 minutes to remove debris, and at 120,000 g for 70 minutes to collect the exosomes. The exosomal pellets were then resuspended in sterile PBS for subsequent experiments [21,22]. The isolated exosomes were observed under a transmission electron microscope (TEM, HITACHI, Japan) to analyze their morphology. Nanoparticle tracking analysis (NTA, Particle Metrix, PMX-120, Germany) was used to confirm the size distribution and concentration of exosomes. Western blotting was performed to detect the levels of exosome markers. HucMSC-Exos can be stored at 4°C for 1 week or at −80°C for up to 3 months.

## Flow cytometric analysis

In this study, hucMSCs were analyzed for surface markers using flow cytometry. Single-cell suspensions were centrifuged, washed with PBS, and stained with PE-conjugated antibodies (CD73/CD90/CD105 for positivity, CD34/CD45 as negative controls) alongside isotype-matched controls. Cells were incubated at 4°C for 30 min in darkness, washed thrice, and resuspended in 1% BSA-PBS. Samples were filtered (40 μm) and analyzed by flow cytometry, gating viable cells via FSC/SSC. A minimum of 10,000 events per sample were recorded, with data processed using FlowJo V10.

## Experimental animal design and handling

Ten-week-old male Sprague-Dawley (SD) rats (average weight 190–210 g) were purchased from Beijing Huafukang Biotechnology Co., Ltd. (Beijing, China) and housed in the SPF-certified animal facility at Gansu University of Chinese Medicine. The rats had free access to water and were fed standard pellets. The dosing regimen and schedule for exosomes and ionizing radiation (IR) are provided in the figure. In subsequent experiments, the rats were randomly divided into three groups: (1) control group; (2) IR group; (3) IR+Exos group. Male SD rats were administered hucMSC-Exos at a dosage of $1.5 \times 10^{11}$ particles/mL (1 mL/kg) via tail vein intravenous injection. The treatment protocol comprised once-daily administrations over three consecutive days, initiated 24 hours prior to abdominal IR and continued for 72 hours post-IR exposure. Experimental rats were subjected to tissue collection on day 5 post-irradiation. Briefly, preoperative anesthesia was induced via intraperitoneal injection (IP) of ketamine (0.2 mL/kg), followed by deep anesthesia with 0.25% tribromoethanol (300 mg/kg, IP). Tissue collection was subsequently performed. Upon completion of the procedure, euthanasia was administered through IP of 2% sodium pentobarbital (200 mg/kg).

## Ionizing radiation protocol

All experiments were performed using the X-RAD 225 (Precision X-ray, Connecticut, USA). In the relevant experiments, cells and rats were irradiated with X-rays at a dose rate of 2.0 Gy/min before and after exosome treatment. IEC-6 cells were seeded at $1 \times 10^6$ per 25 cm² culture flask and irradiated with 10 Gy when cell confluence reached 80–90%. Several SD rats were anesthetized using 0.25% tribromoethanol(200 mg/kg) and irradiated with 12 Gy. The irradiation targeted a 4 cm abdominal area from the xiphoid process to the pubic symphysis covering the gastrointestinal tract, while other body parts were shielded with a 5 cm thick lead block. The radiation dosage for animals was based on previous studies [23,24].

## Histopathologic staining and intestinal biomarker assays

Paraffin-embedded tissue sections (3–5 µm) were deparaffinized and hydrated with xylene, followed by staining with hematoxylin and eosin (H&E) and periodic acid-Schiff (PAS), and finally mounted for observation. We then measured the length of the intestinal villi in SD rats and analyzed the number of intestinal crypts and goblet cells using histopathological staining, as described in previous research [25]. Briefly, intestinal villi are finger-like projections of the mucosal epithelium that extend into the intestinal lumen. Viable crypts are defined as those surrounded by 10 or more irregular, thick-walled non-Paneth cells within a lumen. Goblet cells, rich in mucin and other polysaccharides, stain purple-red after PAS staining, allowing for easy differentiation from other cell types under the optical microscope.

## Immunohistochemical staining

For immunohistochemical staining, tissue sections were deparaffinized and hydrated as described for H&E staining, followed by antigen retrieval by heating the sections in 10 mmol/L citrate buffer at high temperature for 10–15 minutes, and incubation in 3% hydrogen peroxide solution for 15 minutes to block endogenous peroxidase activity. Next, the sections were incubated with serum for 1 hour at room temperature to reduce nonspecific antigen binding, followed by overnight incubation with anti-Ki67 antibody at 4°C. The sections were then incubated with secondary antibody for 30 minutes at 37°C, followed by detection of positive-stained cells using a DAB kit. Finally, the sections were mounted and observed under a optical microscope (Olympus, Japan).

## Detection of oxidative stress markers

After preparing the intestinal homogenate (lysate containing a protease inhibitor) or cell lysis, the supernatant was collected by centrifugation at 4°C and 12,000 × g for 10 minutes. Protein concentrations were quantified using a bicinchoninic acid (BCA) assay. SOD activity was assessed by adding xanthine oxidase and WST-1 substrate according to the manufacturer's instructions, followed by incubation at 37°C for 20 minutes. Absorbance was measured at 450 nm using a

Microplate Reader, and activity was calculated based on the inhibition rate. MDA content was measured by boiling the supernatant with TBA reagent in water for 60 minutes. The mixture was then centrifuged in an ice bath, and absorbance was measured at 532 nm. The MDA concentration was determined based on the standard curve. GSH-Px activity was assessed by adding NADPH and glutathione reductase, monitoring the absorbance decline at 340 nm, and calculating GSH consumption per unit time. The experiment was repeated at least three times for each measurement.

### Cell viability assay

Initially, $1 \times 10^5$ cells were seeded into 96-well plates, and after the cells adhered, their proliferative activity was measured under various radiation doses (0Gy, 1Gy, 2Gy, 5Gy, 10Gy, 15Gy, and 20Gy) to determine the optimal radiation dose (the dose causing 50% cell death). The optimal dose was then fixed, and different concentrations of hucMSC-Exos were added to further determine the ideal concentration of hucMSC-Exos. The proliferative activity of the cells was determined again after transfection. Briefly, 24 hours post-treatment, 10 µL of CCK-8 reagent was added to each well, and the plates were incubated in the dark for 1–2 hours. Absorbance was then measured at 450 nm using a Microplate Reader.

### Measurement of cellular and tissue ROS

After seeding the cells in 6-well plates, ROS levels were assessed 6 hours after IR and hucMSC-Exos treatment using the DCFH-DA fluorescence probe to detect intracellular ROS levels, followed by nuclear staining with Hoechst 33342. For the small intestine tissue, fresh samples were rapidly frozen, embedded, and sectioned into 6 µm thick frozen slices, followed by loading with DCFH-DA probe and DAPI counterstaining. The cells and tissues were incubated with 10 µM DCFH-DA at 37°C in the dark for 30 minutes, and then observed for DCFH fluorescence under a confocal microscope. Fluorescence intensity was analyzed using ImageJ software.

### Cell cloning assay

For the colony formation assay, transfected IEC-6 cells were seeded into 6-well plates at a density of 800 cells per well. Following 24 hours of incubation, cells were exposed to 10 Gy ionizing radiation and subsequently treated with 100 µg/mL hucMSC-Exos. After 24 hours of exosome exposure, the culture medium was replaced with fresh complete medium. Cells were maintained under standard culture conditions (37°C, 5% $CO_2$) for 7–14 days to allow colony formation. Colonies were fixed with 4% paraformaldehyde, stained with 0.5% crystal violet solution, and quantified microscopically. Only well-defined colonies containing ≥50 viable cells were counted for statistical analysis.

### RNA interference

SiRNA was used to silence Nrf2 (si-Nrf2), with the following si-Nrf2 sequences:

Si-Nrf2–1,5 ' -CGAGAAGUGUUUGACUUUAUT-3 ';

Si-Nrf2–2,5 ' -CCGAGUUACAGUGUCUUAAUA-3 ';

Si-Nrf2–3,5 ' -GGAAGUCUUCAGCAUGUUAUT-3 '。

Following the manufacturer's instructions, cells were collected and seeded into 6-well, 24-well, or 96-well plates. After 24 hours, siRNA was transfected using RNAFit RNA-specific transfection reagent. After confirming high silencing efficiency, IEC-6 cells were used for subsequent experiments.

### Quantitative real time polymerase chain reaction (qRT-PCR)

Total RNA was extracted using the M5 Universal RNA Mini Kit according to the manufacturer's instructions. Reverse transcription was performed using the M5 Sprint qPCR RT Kit with genomic DNA (gDNA) remover. Quantitative PCR was

 

conducted using 2X M5 Hiper SYBR Premix Estaq, with β-actin used as the internal reference gene. Gene expression was quantified using the $2^{-\Delta\Delta Ct}$ method. The qRT-PCR primer sequences are listed (S1 Table).

### Western blotting

Tissues or cells were lysed in RIPA lysis buffer containing 1% protease inhibitor cocktail and 1% phosphatase inhibitor cocktail. The lysates were collected and centrifuged at 12,000 rpm for 30 minutes at 4°C. Protein concentration was determined using the BCA assay kit. Equal amounts of protein (30 μg) from each sample were loaded and separated on a 10% SDS-PAGE gel, then transferred to a PVDF membrane. The membrane was blocked with 5% non-fat milk at room temperature for 1 hour, followed by incubation with the corresponding primary antibodies (Nrf2, HO-1, NQO1) overnight at 4°C. After washing, the membrane was incubated with the corresponding secondary antibody at room temperature for 2 hours. The bands were visualized using a gel imaging system, and the grayscale intensity was analyzed using Image J.

### Statistical analysis

Data analysis was performed using GraphPad Prism 8.0 (GraphPad Software, Inc., La Jolla, CA, USA). Survival curves were constructed using the Kaplan-Meier method. Comparisons between two groups were made using an independent samples t-test. For comparisons among multiple groups, one-way analysis of variance (ANOVA) followed by Tukey's multiple comparisons test was used. Statistical significance was defined as: $*p < 0.05$, $**p < 0.01$, $***p < 0.001$, $****p < 0.0001$. Each experiment was performed in triplicate.

## Results

### Culture and identification of hucMSCs, and extraction and characterization of hucMSC-Exos

Under a microscope, hucMSCs exhibited a spindle-shaped morphology (Fig 1A). Flow cytometry identified the presence of hucMSCs specific markers, CD73, CD90, and CD105 as positive (Fig 1B-D), while hematopoietic stem cell markers CD34 and CD45 were negative (Fig 1E-F). TEM revealed that the exosomes displayed a typical oval shape (Fig 1G-H). NTA indicated an average diameter of approximately 116 nm, with the exosome concentration initially at $1.5 \times 10^{11}$ particles/mL (Fig 1I). The size distribution of the exosomes is shown in (Fig 1J) . Western blot analysis confirmed the presence of exosomal markers CD9, CD63, CD81, and TSG101, indicating that the vesicles were exosomes (Fig 1K). Calnexin was negative, suggesting that the exosomes were not contaminated with endoplasmic reticulum components during the purification process (Fig 1K).

### HucMSC-Exos treatment reconstructs the intestinal mucosal barrier and improves survival rate

To investigate the role of exosomes in radiation-induced intestinal injury in rats, we administered 1 mL/kg of hucMSC-Exos before and after irradiation and examined the morphological changes of the rats' intestines on day 5 post-12 Gy IR exposure (Fig 2A). Our results showed that the small intestines of rats exposed to IR exhibited ischemic necrosis and watery diarrhea. However, after continuous hucMSC-Exos treatment, these symptoms were significantly alleviated, and intestinal blood flow improved (Fig 2B). Similarly, hucMSC-Exos treatment notably increased the colon length in rats following IR (Fig 2C, D). H&E staining revealed that, compared to the control group, rats in the IR group had significantly shortened villi, some of which were detached and necrotic, with a marked reduction in crypt depth. In contrast, hucMSC-Exos treatment alleviated these changes (Fig 2E, G, H). Additionally, PAS staining showed a notable reduction in goblet cells in the small intestine epithelium after IR exposure; however, hucMSC-Exos treatment resulted in an increase in the number of goblet cells (Fig 2F, I). We also monitored and analyzed the weight changes of the rats over the course of the study, finding that hucMSC-Exos treatment significantly increased the body weight of IR-exposed rats, enhancing the efficiency of intestinal absorption (Fig 2J). Survival analysis further demonstrated that hucMSC-Exos treatment significantly improved the survival rate of rats post-IR exposure (Fig 2K).

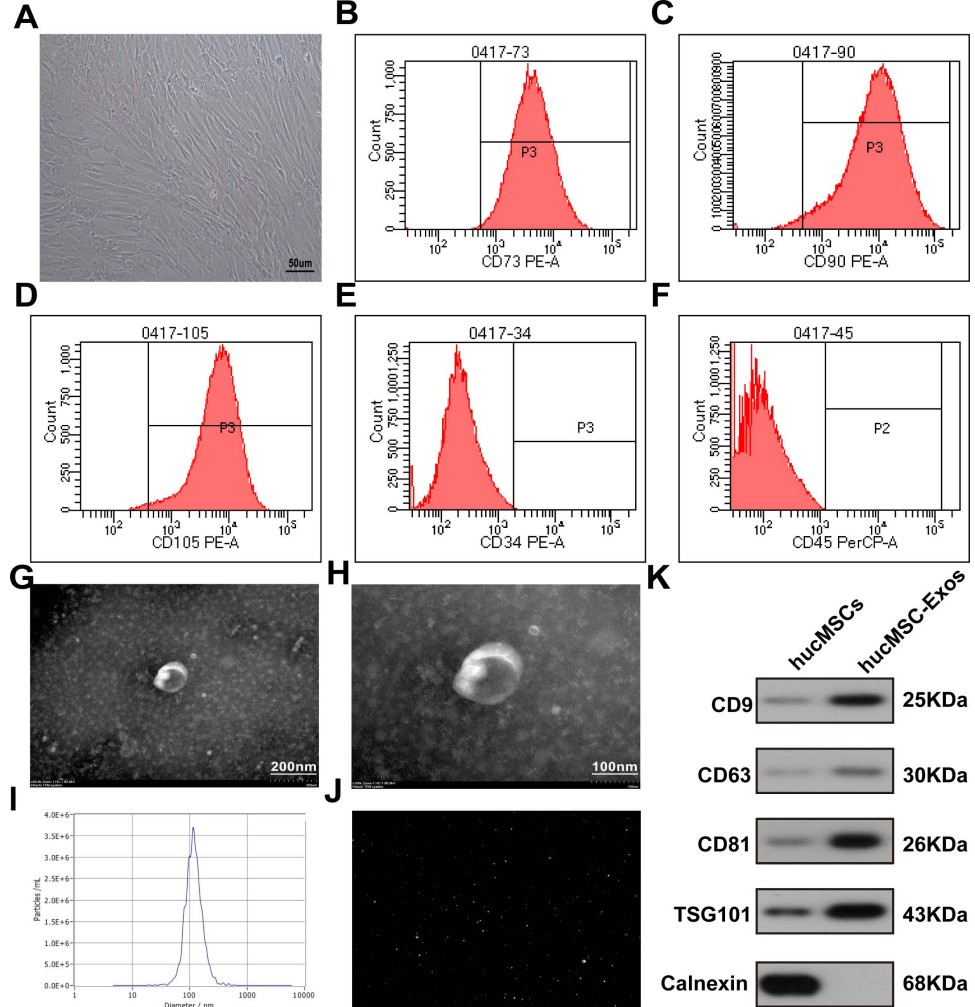

**Fig 1. Identification of hucMSCs and exosomes.** (A) hucMSCs exhibited a spindle shape microscopically. Scale bar: 50 μm. (B-F) Flow cytometric analysis of stem cell markers CD73, CD90, and CD105 was positive, while hematopoietic stem cell markers CD34 and CD45 were negative. (G-H) Transmission electron microscopy(TEM) was used to examine the morphology of hucMSC-Exos. (I-J) Nanoparticle tracking analysis (NTA) was performed to determine the diameter, concentration, and particle size distribution of hucMSC-Exos. (K) Western blot analysis was used to detect exosome-related markers in hucMSC-Exos and hucMSCs.

### HucMSC-Exos promote rat intestinal epithelial proliferation and reduce oxidative stress

To assess the effect of exosomes on intestinal epithelial cell proliferation, we performed immunohistochemical staining to detect Ki67＋cells. Our results demonstrated that, compared to the IR group, rats treated with hucMSC-Exos exhibited a significant increase in the number of Ki67＋cells after IR exposure (Fig 3A,B). We then investigated the ability of hucMSC-Exos to alleviate oxidative stress. Using a fluorescent probe, we analyzed the fluorescence intensity of ROS in the small intestinal epithelial tissue. The results showed a significant increase in ROS fluorescence in the intestinal epithelium of rats after IR, whereas hucMSC-Exos treatment significantly reduced the fluorescence intensity (Fig 3C,D). Additionally, we measured classic redox markers such as SOD, MDA, and GSH-Px using Biochemical assay kits. The results indicated that hucMSC-Exos significantly reduced oxidative stress damage induced by IR (Fig 3E-G). Given the pivotal role of the Nrf2 signaling pathway in maintaining redox balance and protein homeostasis [26], we investigated whether hucMSC-Exos could modulate the Nrf2/HO-1/NQO1 pathway. mRNA and protein level analyses revealed that IR

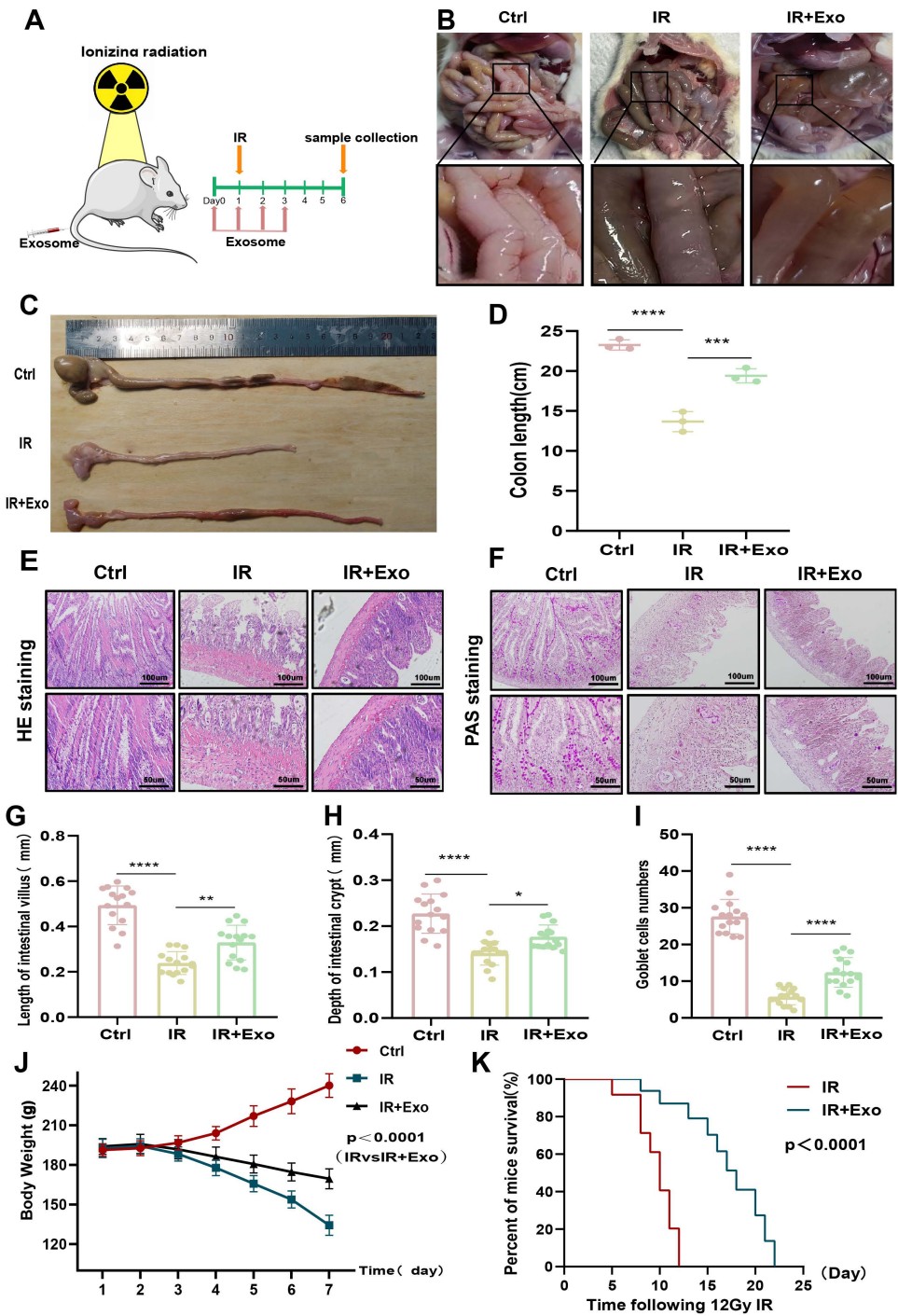

**Fig 2. HucMSC-Exos remodel the intestinal mucosal barrier and improves survival.** (A) Rats were administered exosomes both before and after irradiation, and intestinal segments were collected on day 5 after irradiation, as shown in the figure (n = 6 animals/group). (B) Representative images of intestinal conditions in each group. (C-D) Colonic lengths in each group. (E) Hematoxylin and eosin (H&E) staining showing representative images of the cross-sectional structure of the small intestine. Magnification: ×200 (top), ×400 (bottom). (F) Representative image of periodic acid-Schiff (PAS) staining. Magnification: ×200 (top), ×400 (bottom). (G-I) Length of small intestinal villi, depth of crypts, and number of cup cells based on H&E and PAS staining analysis. (J) Changes in body weight in rats of different groups (n = 6/group). (K) Kaplan-Meier survival analysis of rats following 12 Gy irradiation (n = 10/group).The data are representative of three independent experiments; one-way ANOVA (Tukey's multiple comparisons test) ; *$p < 0.05$; **$p < 0.01$, ***$p < 0.001$, ****$p < 0.0001$.

stimulates the rat's endogenous antioxidant response, and hucMSC-Exos treatment further amplified this effect, thereby mitigating oxidative damage (Fig 3H-L).

## HucMSC-Exos alleviate oxidative stress in IEC-6 Cells

We further investigated the antioxidant effects of hucMSC-Exos in IR-induced IEC-6 cells (rat intestinal epithelial cells) in vitro. First, we used a CCK8 assay to determine the optimal radiation dose and exosome concentration (Fig. 4A,B), thus establishing the in vitro radiation model. Next, we measured ROS fluorescence and the expression of redox markers, including SOD, MDA, and GSH-Px, in IEC-6 cells. The results demonstrated that hucMSC-Exos significantly reduced ROS fluorescence (Fig 4C,D) and effectively alleviated oxidative stress (x). Given the critical role of the Nrf2 pathway in redox regulation, we employed qRT-PCR and Western blotting to evaluate the expression of Nrf2, HO-1, and NQO1. The findings were consistent with the in vivo experiments, showing that exosomes enhanced Nrf2 pathway expression and mitigated oxidative damage in IEC-6 cells (Fig 4H-L).

## HucMSC-Exos mitigate IR-induced oxidative damage and enhance cellular proliferative capacity by activating the Nrf2/HO-1/NQO1 antioxidant signaling pathway

To further elucidate the functional linkage between hucMSC-Exos and the canonical Nrf2/HO-1/NQO1 oxidative stress pathway, we performed Nrf2 gene silencing (si-Nrf2) in irradiated IEC-6 cells. Silencing efficiency was rigorously validated through qRT-PCR and Western blotting (Fig 5A-C), with si-Nrf2–1 (highest knockdown efficiency) selected for subsequent experiments.

Colony formation and CCK-8 assays revealed that hucMSC-Exos treatment significantly restored proliferation in irradiated cells, whereas si-Nrf2 transfection markedly attenuated this rescue effect ($p < 0.0001$, si-Nrf2 group vs si-NC group), indicating Nrf2-dependent regulation of proliferative recovery (Fig 5D-F). Mechanistically, qRT-PCR analysis demonstrated that Nrf2 knockdown abolished radiation-induced upregulation of downstream effectors HO-1 and NQO1. Crucially, hucMSC-Exos failed to activate HO-1/NQO1 expression in si-Nrf2-treated cells ($p < 0.0001$, si-Nrf2 + Exos+IR group vs si-NC+Exos+IR group), confirming pathway dependency (Fig 5G,H). Western blotting further corroborated these findings, showing that si-Nrf2 suppressed hucMSC-Exos-mediated enhancement of antioxidant protein levels ($p < 0.0001$)(Fig 5I-K). Collectively, these data demonstrate that hucMSC-Exos counteract radiation-induced oxidative damage and stimulate proliferation via direct activation of the Nrf2/HO-1/NQO1 signaling axis.

## Discussion

Radiotherapy plays a pivotal role in the comprehensive treatment of gastrointestinal and pelvic malignancies. However, it may lead to radiation-induced intestinal injury (RII) in certain patients. Mild cases commonly present with diarrhea, abdominal pain, mucus stools, and slight bleeding, whereas severe cases may result in more serious complications, including intestinal obstruction and fistulas, which are challenging to treat and significantly impair the patients' quality of life [2,3]. Currently, there is no universally recognized optimal treatment for acute radiation-induced intestinal injury (ARII), and existing therapeutic approaches often produce suboptimal outcomes. Thus, understanding the mechanisms underlying radiation-induced intestinal damage and developing effective strategies to alleviate such injuries remain critical. Exosomes, small membrane-bound nanoparticles [27,28], have been shown in numerous studies to enhance intercellular communication, facilitate waste clearance, and promote cell repair and regeneration [27,29,30]. However, research on the role of exosomes in ARII is limited. This study focuses on hucMSC-Exos, exploring their involvement in ARII and assessing their therapeutic potential.

In this study, we hypothesized that hucMSC-Exos play a crucial role in ARII in rats. Based on this hypothesis, we comprehensively validated the protective effects of hucMSC-Exos against radiation-induced damage. We determined the appropriate radiation dose, exosome concentration, and administration route for the rat ARII model, based on previous

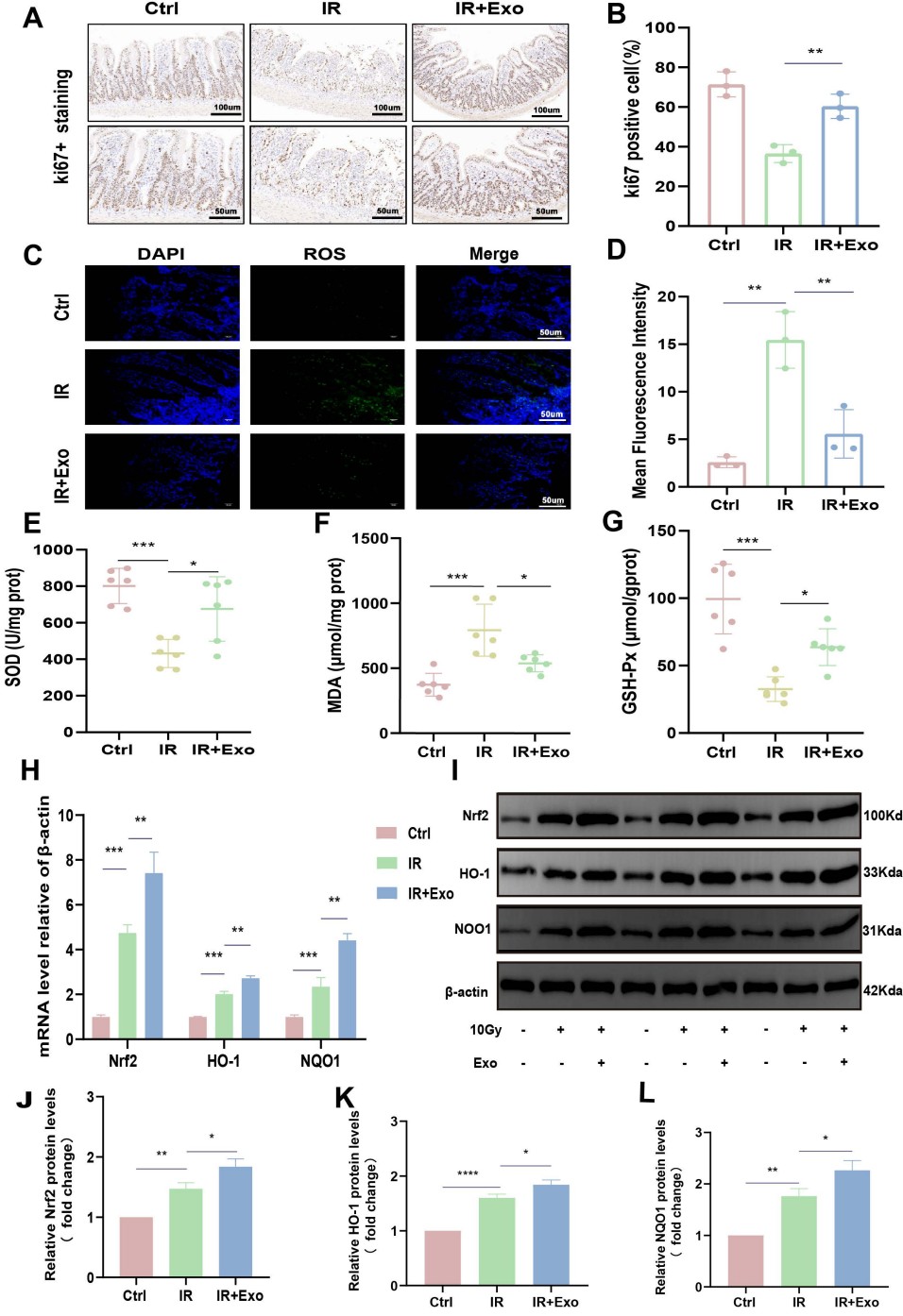

**Fig 3. HucMSC-Exos promote proliferation and reduces oxidative stress injury in the rat intestinal epithelium.** (A-B) Immunohistochemistry (IHC) staining showing representative images of Ki67 + expression in a cross-section of the small intestine on day 5 after irradiation, along with analysis of the positive rate. Magnification: ×200 (top), ×400 (bottom). (C-D) Expression of reactive oxygen species (ROS) in the epithelial tissue of the rat small intestine and analysis of fluorescence intensity. Scale bar: 100 μm. (E-G) Biochemical assay kits was performed to detect the expression of SOD, GSH-Px, and MDA redox markers in the tissues. (H) qRT-PCR was performed to detect the expression levels of Nrf2, HO-1, and NQO1 in small intestinal tissues. (I) Western blot analysis of Nrf2, HO-1, and NQO1 expression in small intestinal tissues. (J-L) Relative quantitative analysis of Nrf2, HO-1, and NQO1 protein levels in small intestinal tissues. The data are representative of three independent experiments; one-way ANOVA (Tukey's multiple comparisons test) ; *$p < 0.05$; **$p < 0.01$, ***$p < 0.001$, ****$p < 0.0001$.

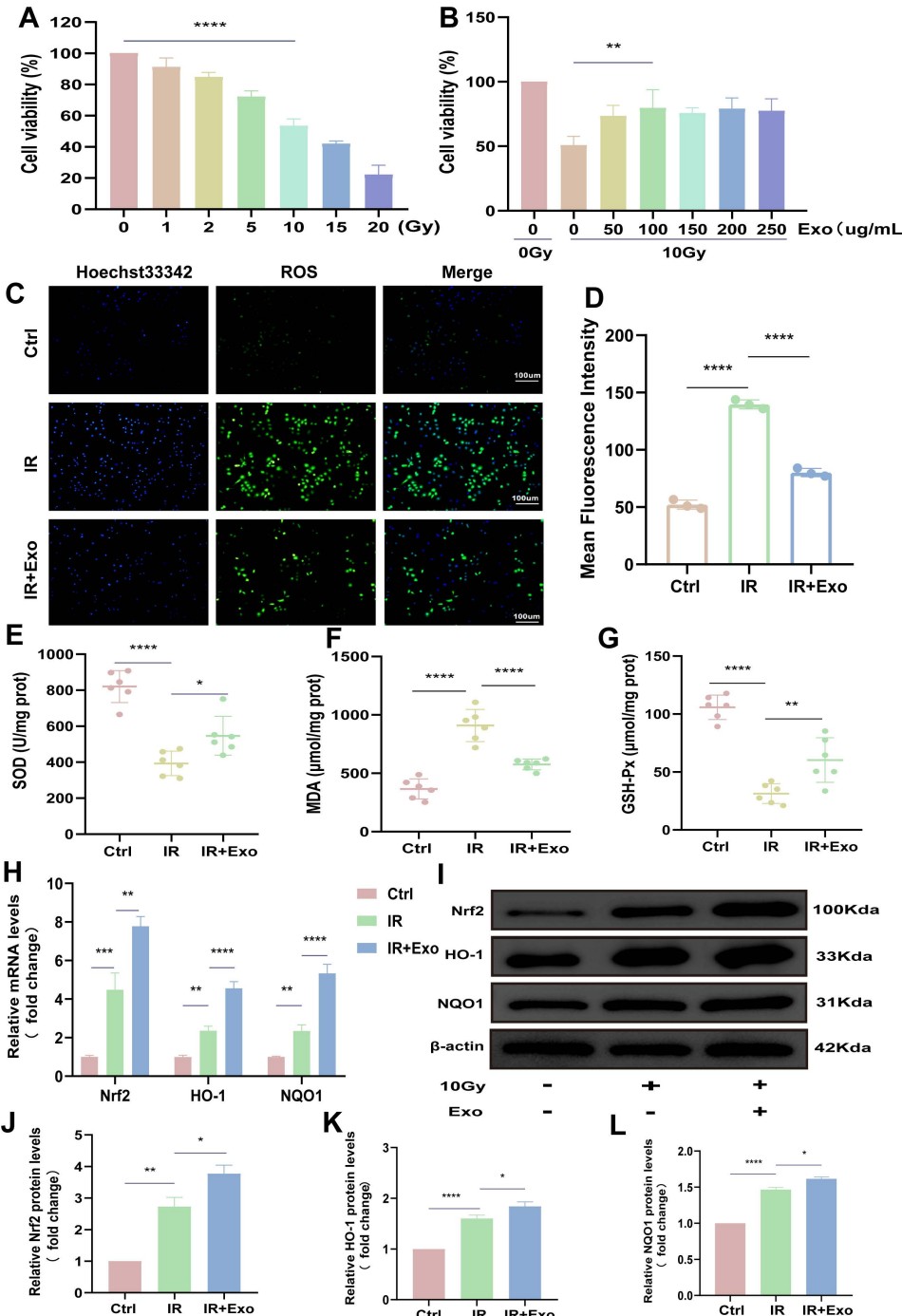

**Fig 4. HucMSC-Exos alleviate oxidative stress injury in IEC-6 cells.** (A) The effects of different radiation doses on IEC-6 cell activity were assessed using the CCK-8 assay to determine the optimal radiation dose. (B) The CCK-8 assay was used to determine the optimal exosome dose under a 10 Gy irradiation dose. (C-D) Representative fluorescence images and fluorescence intensity analysis of ROS in different groups of IEC-6 cells. (E-G) Biochemical assay kits for the expression of SOD, GSH-Px, and MDA redox markers in IEC-6 cells. (**H**) qRT-PCR to detect the expression levels of Nrf2, HO-1, and NQO1 in IEC-6 cells. (I) Western blot analysis of Nrf2, HO-1, and NQO1 expression in IEC-6 cells. (J-L) Relative quantitative analysis of Nrf2, HO-1, and NQO1 protein levels in IEC-6 cells.The data are representative of three independent experiments; one-way ANOVA (Tukey's multiple comparisons test) ; *$p < 0.05$; **$p < 0.01$. ***$p < 0.001$, ****$p < 0.0001$.

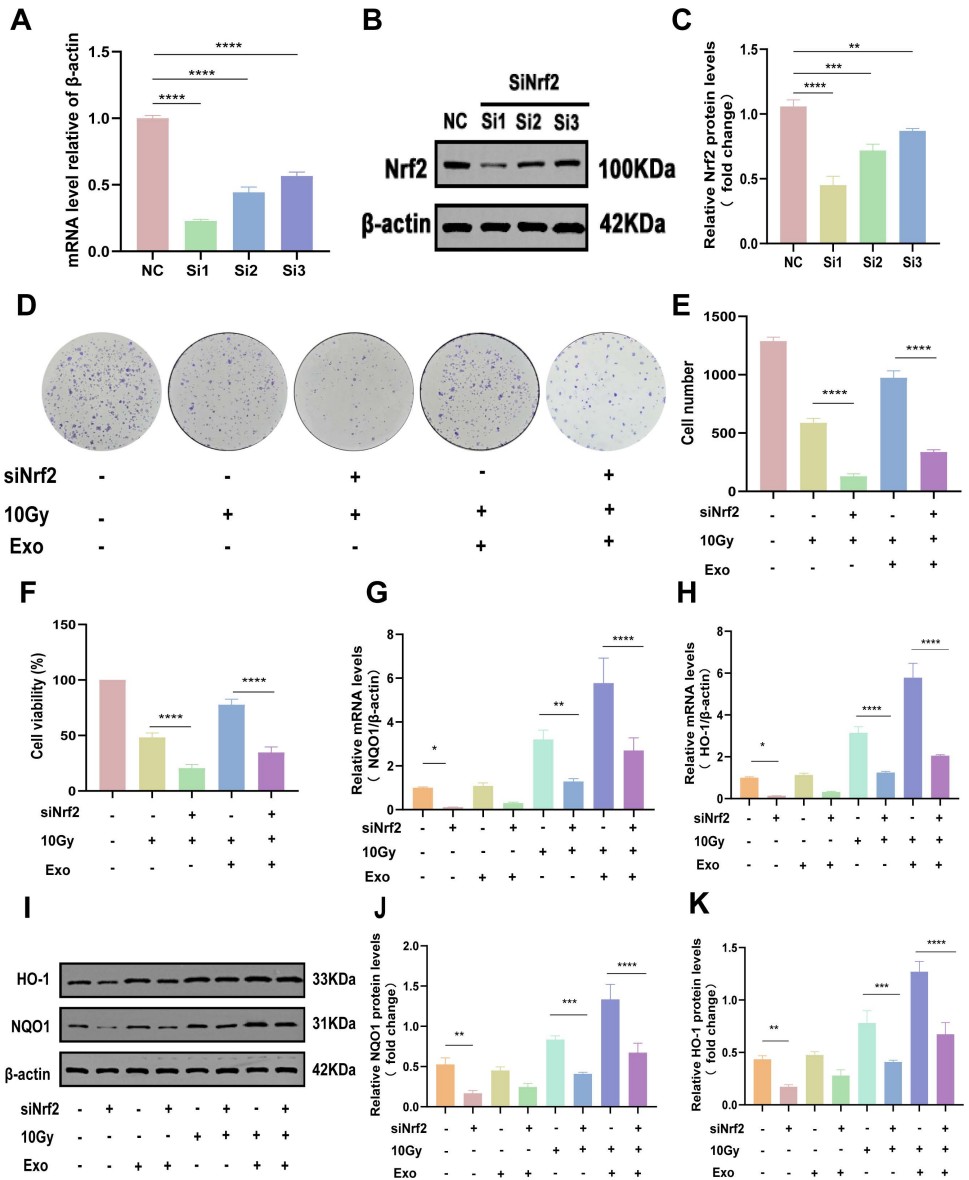

**Fig 5. HucMSC-Exos mitigate IR-induced oxidative damage and enhance cellular proliferative capacity by activating the Nrf2/HO-1/NQO1 anti-oxidant signaling pathway.** We silenced the Nrf2 transcription factor using siRNA and verified and analyzed the silencing efficiency through qRT-PCR (**A**), and Western blot analysis (**B and C**) to identify the optimal knockdown strand. Colony formation assay (**D-E**) and CCK-8 assay (**F**) were performed to assess the impact of hucMSC-Exos treatment on the proliferation of transfected cells. (**G-H**) qRT-PCR analysis to compare the expression levels of HO-1 and NQO1 before and after silencing. (I) Western blot analysis to compare the expression levels of HO-1 and NQO1. (J-K) Relative quantitative analysis of HO-1 and NQO1 protein levels before and after silencing.The data are representative of three independent experiments; one-way ANOVA (Tukey's multiple comparisons test) ; *$p < 0.05$; **$p < 0.01$. ***$p < 0.001$, ****$p < 0.0001$.

studies and preliminary experiments [23,24,31]. Subsequently, we evaluated the general condition of the rats' intestines. Our results showed that hucMSC-Exos significantly alleviated radiation-induced ischemic necrosis and diarrhea, and, compared to radiation alone, prolonged the colon length in the rats. Furthermore, ionizing radiation caused varying degrees of intestinal villus blunting, fusion, and loss, along with a reduction in crypt depth, disrupting epithelial integrity. Goblet cells, responsible for secreting mucins that protect and lubricate the mucosal surface, play a vital role in

maintaining the intestinal mucosal barrier and regulating epithelial turnover [25]. Interestingly, histopathological examination of the intestinal tissue revealed that the length of the intestinal mucosa, crypt depth, and goblet cell count were significantly restored in rats treated with hucMSC-Exos, and survival rates were prolonged. These findings suggest that hucMSC-Exos may protect against radiation-induced ARII by remodeling the intestinal mucosal barrier. Ki67, a classical biomarker of cell proliferation, is a nuclear protein strictly associated with cell cycle progression [32]. It is exclusively expressed in actively proliferating cells (G1, S, G2, and M phases) and regulates mitosis by interacting with nucleolar components and chromatin [33]. Immunohistochemical analysis in this study revealed that hucMSC-Exos treatment significantly increased the proportion of Ki67 + intestinal epithelial cells in radiation-injured rats, indicating that hucMSC-Exos promote epithelial regeneration by facilitating cell cycle progression. This increased proliferative activity not only supports the self-repair of the intestinal epithelium following radiation exposure but, more importantly, facilitates the structural and functional restoration of the intestinal mucosal barrier by promoting tight junction protein reconstruction and enhancing mucus secretion.

Oxidative stress occurs when the production of reactive oxygen species (ROS) and other oxidants exceeds the capacity of the antioxidant system (including endogenous enzymes and small-molecule antioxidants), resulting in a redox imbalance. Ionizing radiation has been shown to directly break down water molecules, generating highly reactive free radicals, including hydroxyl radicals ($\cdot$OH), superoxide anions ($\cdot O_2^-$), and peroxide ions ($\cdot O_2^{2-}$), which leads to the accumulation of ROS [34,35]. Our data indicate that hucMSC-Exos treatment significantly reduced ROS levels in the intestinal epithelium of irradiated rats and promoted the expression of superoxide dismutase (SOD) and glutathione peroxidase (GSH-Px), while effectively inhibiting the activity of lipid peroxidation product malondialdehyde (MDA). These results suggest that the high antioxidant activity of hucMSC-Exos may contribute to the treatment of ARII. Additionally, the Keap1-Nrf2 pathway, a key antioxidant defense mechanism [16], regulates the expression of various antioxidant enzymes, such as HO-1, NQO1, SOD, and GSH-Px [26,36]. While Nrf2 is known to play a prominent role in the liver [35,37], brain [38], lungs [39], skin [40] and so on, its role in the intestine is less frequently studied [16]. Therefore, we examined the expression of Nrf2 and its downstream transcription factors, HO-1 and NQO1, in the rat intestine. Our findings suggest that hucMSC-Exos may enhance Nrf2 nuclear translocation and promote the expression of antioxidant proteins and enzymes.

To investigate the role of hucMSC-Exos in mitigating oxidative damage in intestinal epithelial cells, an in vitro experimental model was established using rat IEC-6 cells. Initially, an in vitro radiation model was successfully constructed by determining the optimal radiation dose and exosome concentration via a CCK8 assay. Subsequently, analysis of ROS levels and redox markers (SOD, MDA, and GSH-Px) in IEC-6 cells revealed that the results closely mirrored those from animal studies, thereby indirectly validating the reliability of this in vitro model for simulating radiation-induced oxidative damage. Notably, mechanistic studies demonstrated that the Nrf2 signaling pathway plays a central regulatory role in this process. Western blot analysis indicated that hucMSC-Exos synergistically enhanced radiation-induced Nrf2 expression and significantly upregulated its downstream target genes, HO-1 and NQO1. To further confirm the necessity of this signaling pathway, siRNA technology was employed to knock down Nrf2 (achieving a silencing efficiency of over 80%), thereby generating a loss-of-function model. The results demonstrated that Nrf2 deletion significantly abrogated the pro-proliferative effect of hucMSC-Exos and eliminated its regulatory impact on HO-1 and NQO1 expression; this gene-phenotype co-regulation was corroborated by both qRT-PCR and protein blot analyses (Fig 5G-K).

In this study, we provide the first evidence that hucMSC-derived exosomes restore redox homeostasis in radiation-exposed intestinal epithelial cells through dual mechanisms mediated by the Nrf2/HO-1/NQO1 signaling axis: (1) direct scavenging of radiation-induced excessive ROS, and (2) epigenetic enhancement of endogenous antioxidant defense systems. These findings establish a theoretical foundation for developing exosome-based therapeutics against ARII. Unlike conventional antioxidants (e.g., amifostine), exosomes exhibit inherent tissue tropism and superior biosafety. Their bioactive cargo (including miRNAs, lncRNAs, circRNAs, and functional proteins) enables precise

modulation of intestinal oxidative microenvironments while circumventing systemic side effects associated with conventional drug delivery [41]. Recent advances in engineered exosomes have demonstrated enhanced lesion-specific accumulation and therapeutic stability, significantly improving their clinical translation potential. For instance, modified MSC exosomes (SARS-CoV-2-S-RBD/miR-486-5p-engineered MSC-Exos) have shown dual efficacy in suppressing ferroptosis and fibrosis in pulmonary epithelial cells in vitro, along with mitigating radiation-induced lung injury in ACE2-humanized murine models [42].

This study acknowledges three potential limitations: First, in vitro validation of oxidative pathways using intestinal epithelial cell models may not fully recapitulate the physiological complexity of in vivo systems. Second, despite emerging evidence on exosome-mediated radioprotection, the identity and functional hierarchy of specific exosomal non-coding RNAs (e.g., miRNAs, lncRNAs) and their target genes in mitigating ARII remain to be systematically elucidated. Third,-clinical validation of exosome therapy for RII remains scarce. We will focus on these areas in our follow-up investigations. Future studies should elucidate the specific components within exosomes that regulate ARII and prioritize the development of advanced intestinal organoid models capable of recapitulating human pathological microenvironments. This approach aims to bridge the translational gap between exosome research and personalized therapies, while simultaneously advancing multicenter preclinical trials to systematically evaluate long-term biosafety profiles and explore synergistic strategies with current radiotherapy regimens. As multifunctional vectors in regenerative medicine, exosomes hold promise for evolving from mechanistic exploration to precision interventions. This cell-free therapeutic paradigm exhibits transformative potential in managing abdominopelvic radiation-induced enteropathy, potentially reshaping the clinical landscape of ARII.

In summary, our study elucidates the therapeutic effects and specific mechanisms of hucMSC-Exos in ARII, highlighting their antioxidant role (Fig 6), and provides a novel treatment strategy for combating radiation-induced clinical intestinal damage.

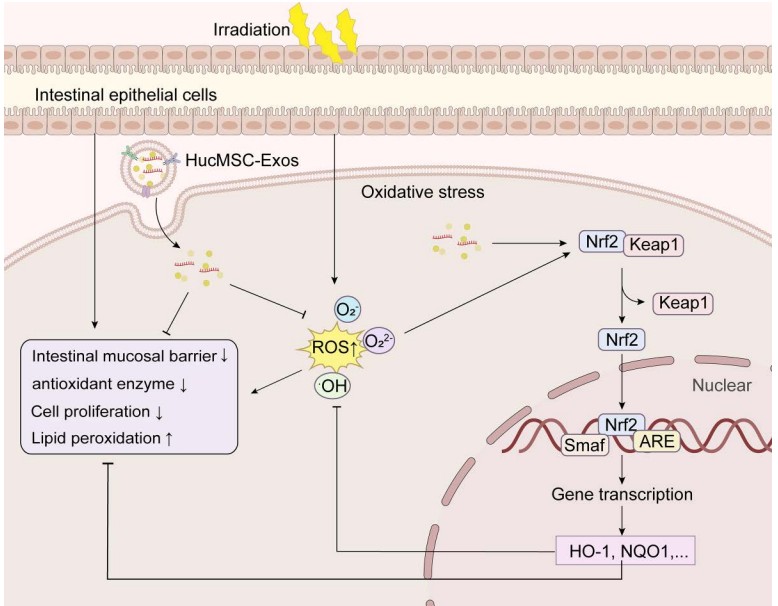

**Fig 6. The antioxidant mechanism of hUCMSC-Exos in ARII.** HucMSC-Exos alleviate radiation-induced acute oxidative damage by inhibiting ROS accumulation, reducing lipid peroxidation, and further amplifying the Nrf2/HO-1/NQO1 signaling pathway.

## Conclusions

In conclusion, our findings demonstrate that hucMSC-Exos enhance survival and functional integrity of IR-exposed intestinal epithelial cells in both in vitro and in vivo models by alleviating oxidative injury through Nrf2/HO-1/NQO1 pathway activation, positioning them as a novel cell-free therapy for ARII. The broader potential of engineered exosomes in regenerative medicine, including targeted drug delivery remains a promising frontier.

## Supporting information

**S1 Raw images. Raw images of Western blot (for Fig 1k, Fig 3I, Fig 4I, Fig 5B, Fig 5I).**
(PDF)

**S1 Table. PCR primers used in this study.**
(DOCX)

**S1 Data. Minimal Data Set.**
(ZIP)

## Acknowledgments

I would like to express my gratitude to the General Surgery Department I of Gansu Provincial Central Hospital for their support and encouragement, as well as to Ms. JJW for her invaluable assistance.All authors declare that they have not use AI-generated work in this manuscript.

## Author contributions

**Conceptualization:** Weikai Chen.

**Formal analysis:** Jinbao Wang, Gaosheng Yang, Yanjie Li.

**Funding acquisition:** Jianping Yu, Xiaopeng Han, Weikai Chen.

**Investigation:** Jinbao Wang, Gaosheng Yang, Yanjie Li.

**Methodology:** Weikai Chen.

**Visualization:** Hongyu Wang.

**Writing – original draft:** Hongyu Wang.

**Writing – review & editing:** Jianping Yu, Xiaopeng Han.

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
