## [Decision Letter · Decision Letter 0]

14 Apr 2025

Dear Dr. Han,

Thank you for submitting your manuscript to PLOS ONE. After careful consideration, we feel that it has merit but does not fully meet PLOS ONE’s publication criteria as it currently stands. Therefore, we invite you to submit a revised version of the manuscript that addresses the points raised during the review process.

We look forward to receiving your revised manuscript.

Kind regards,

Sabata Martino, Ph.D

Academic Editor

PLOS ONE

Journal Requirements:

This work was supported by the Natural Science Foundation of Gansu Province (grant No. 23JRRA1386) and the Lanzhou Science and Technology Bureau (No. 2023-4-65).

Additional Editor Comments :

no comments

Reviewers' comments:

Reviewer's Responses to Questions

**Comments to the Author**

1. Is the manuscript technically sound, and do the data support the conclusions?

Reviewer #1: Yes

Reviewer #2: Yes

2. Has the statistical analysis been performed appropriately and rigorously?

Reviewer #1: Yes

Reviewer #2: Yes

3. Have the authors made all data underlying the findings in their manuscript fully available?

Reviewer #1: Yes

Reviewer #2: Yes

4. Is the manuscript presented in an intelligible fashion and written in standard English?

Reviewer #1: Yes

Reviewer #2: Yes

Reviewer #1: Weak points

Minor weak points

There are simply typos and minor adjustments to be made:

• Line 39; 40; 68; 70; 87; 132; 181; 200; 203; 207; 323; 326; 337; 340; 358; 364; 388; 459; 471; 473 there is no space after the dot;

• Line 168; 211 the number of the cells is not written with exponential, as previously done;

• Line 327; 379; 381; 404; 418; 424; 425; 426; 427; 459; 462 the citation are written at the same level of the text and not as a superscript, as previously done;

• Line 405 Ki67+ have the first letter written in lowercase and not in uppercase, as done before

• The paragraph of the Flow cytometric analysis is written with a different style respect to the other, like a laboratory protocol, I consider to rewrite this paragraph with the same style of the other;

• Line 200 what method was used to determine the protein concentration? Was a standard curve constructed first? If so, how?

• Line 202-203 what instrument was used to measure the absorbance?

• Figure 1A the scale bar is not readable and in Figure 1G and 1H the scale bar is not present;

• Figure 1K which normalizing protein was used? I would insert its image;

• Figure 4C the scale bar are not present

Major weak points

There are no major weak points

Reviewer #2: The text is flowing in reading, there are no inconsistencies but the form of the text should be improved. Also, space between one paragraph and the next should be added.

- Line 158, are present 2 dots

- Line 126, 1x10^5 must be written 1x105

- Line 168, 1x106 must be written 1x106

- Line 185, which type of microscope?

- Line 195, which type of microscope

- Line 203, how did you measure the absorbance?

- Line 211, 1x10^5 must be written 1x105

- Line 219, which instrument is it?

- The style of the references is different in the conclusion than in the rest of the article. Check that all bibliographic references are uniform with each other and adhere to the style that the journal dictates.

- Line 426, commas and References

- Images 1, 2, 3 e 4 pay attention to the scale bar. Is not visible

- Image 3C the words Dapi, Ros and Merge seem to refer to the image above; so bring it closer to the corresponding image.

- “CON” to substitute with “CTRL”

- Unit measures in image 4, graph B

**Do you want your identity to be public for this peer review?** For information about this choice, including consent withdrawal, please see our Privacy Policy

Reviewer #1: No

Reviewer #2: No

---

## [Author Response · Author response to Decision Letter 1]

17 Apr 2025

Dear reviewer

Thank you for taking time to review our manuscript.

We studied your comments and revised our draft accordingly. We also went through our manuscript and made corrections on the texts. Minor changes to the manuscript are highlighted. Hope that our revised draft will meet with your approval. Furthermore� we would like to show the details as follow

Reviewer #1

1.Line 39; 40; 68; 70; 87; 132; 181; 200; 203; 207; 323; 326; 337; 340; 358; 364; 388; 459; 471; 473 there is no space after the dot.

The author's answer:We have corrected these errors in the article.

2.Line 168; 211 the number of the cells is not written with exponential, as previously done.

The author's answer:We have corrected these errors in the article.

3.Line 327; 379; 381; 404; 418; 424; 425; 426; 427; 459; 462 the citation are written at the same level of the text and not as a superscript, as previously done

The author's answer:We have corrected these errors in the article.

4.Line 405 Ki67+ have the first letter written in lowercase and not in uppercase, as done before

The author's answer:We have corrected these errors in the article.

5.The paragraph of the Flow cytometric analysis is written with a different style respect to the other, like a laboratory protocol, I consider to rewrite this paragraph with the same style of the other.

The author's answer:We have revised the sentence in the article. The details as follows: In this study�hucMSCs were analyzed for surface markers using flow cytometry. Single-cell suspensions were centrifuged, washed with PBS, and stained with PE-conjugated antibodies (CD73/CD90/CD105 for positivity, CD34/CD45 as negative controls) alongside isotype-matched controls. Cells were incubated at 4℃ for 30 min in darkness, washed thrice, and resuspended in 1% BSA-PBS. Samples were filtered (40 μm) and analyzed by flow cytometry, gating viable cells via FSC/SSC. A minimum of 10,000 events per sample were recorded, with data processed using FlowJo V10.

6.Line 200 what method was used to determine the protein concentration? Was a standard curve constructed first? If so, how?

The author's answer:We have revised the sentence in the article. The details as follows: Protein concentrations were quantified using a bicinchoninic acid (BCA) assay. Briefly, serially diluted bovine serum albumin (BSA) standards and test samples were incubated with BCA working reagent at 37°C for 30 min. Absorbance at 562 nm was measured using a Microplate Reader (Multiskan FC, thermofisher, USA), with sample concentrations calculated through linear regression analysis of the standard curve.

7.Line 202-203 what instrument was used to measure the absorbance?

The author's answer:We have revised the sentence in the article. The details as follows: Absorbance was measured at 450 nm using a Microplate Reader.

8.Figure 1A the scale bar is not readable and in Figure 1G and 1H the scale bar is not present

The author's answer:We have revised the Figure 1A,G,H.

9.Figure 1K which normalizing protein was used? I would insert its image

The author's answer:Thank you for your question.According to the protein marker identification criteria established by the International Society for Extracellular Vesicles (ISEV), the characterization of extracellular vesicles (EVs) should include detection of at least two transmembrane proteins and one cytosolic protein. Therefore, the standard practice involves analysis of three positive markers (mandatorily including one transmembrane protein and one cytosolic protein) alongside one negative marker. For exosomal positive protein markers, transmembrane proteins such as CD9, CD63, and CD81 can be selected, while cytosolic proteins may include TSG101, Alix, or HSP70. Appropriate negative protein markers for exosome identification comprise endoplasmic reticulum-associated proteins (e.g., Calnexin), nuclear proteins (e.g., Histone 3), or Golgi apparatus proteins (e.g., GM130). Given the qualitative nature of this analysis, detection of loading control proteins is generally omitted in most experimental protocols.

10.Figure 4C the scale bar are not present

The author's answer:We have revised the Figure 4C

Reviewer #2

1.Line 158, are present 2 dots

The author's answer:We have corrected these errors in the article.

2.Line 126, 1x10^5 must be written 1x105

The author's answer:We have corrected these errors in the article.

3.Line 168, 1x106 must be written 1x106

The author's answer:We have corrected these errors in the article.

4.Line 185, which type of microscope?

The author's answer:Goblet cells are typically observed under an optical microscope(Olympus, Japan) following PAS staining.

5.Line 195, which type of microscope

The author's answer:Ki67+ immunohistochemical (IHC) staining sections are observed under an optical microscope (Olympus, Japan).

6.Line 203, how did you measure the absorbance?

The author's answer:We have revised the sentence in the article. The details as follows: Absorbance was measured at 450 nm using a Microplate Reader(Multiskan FC, thermofisher, USA).

7.Line 211, 1x10^5 must be written 1x105

The author's answer:We have corrected these errors in the article.

8.Line 219, which instrument is it?

The author's answer:It is the Microplate Reader

9.The style of the references is different in the conclusion than in the rest of the article. Check that all bibliographic references are uniform with each other and adhere to the style that the journal dictates.

The author's answer:We have corrected these errors in the article.

10.Line 426, commas and References

The author's answer:We have corrected these errors in the article.

11.Images 1, 2, 3 e 4 pay attention to the scale bar. Is not visible

The author's answer:We have corrected these Figures in the article.

12.Image 3C the words Dapi, Ros and Merge seem to refer to the image above; so bring it closer to the corresponding image.

The author's answer:We have corrected the Figure 3C in the article.

13.''CON'' to substitute with ''CTRL''

The author's answer:We have replaced ''CON'' with ''CTRL'' in the article.

14.Unit measures in image 4, graph B

The author's answer:We have corrected the Figure 4B in the article.

Thank you very much for your attention and time. Look forward to hearing from you.

Best wishes,

Xiaopeng Han

17 April , 2025

---

## [Editor Report · Decision Letter 1]

23 Apr 2025

Human umbilical cord mesenchymal stem cell-derived exosomes mitigate acute radiation-induced intestinal oxidative damage via the Nrf2/HO-1/NQO1 signaling pathway

PONE-D-25-12676R1

Dear Dr. Han,

We’re pleased to inform you that your manuscript has been judged scientifically suitable for publication and will be formally accepted for publication once it meets all outstanding technical requirements.

Kind regards,

Sabata Martino, Ph.D

Academic Editor

PLOS ONE

Additional Editor Comments (optional):

no comments
---

## [Editor Report · Acceptance letter]

PONE-D-25-12676R1

PLOS ONE

Dear Dr. Han,

I'm pleased to inform you that your manuscript has been deemed suitable for publication in PLOS ONE. Congratulations! Your manuscript is now being handed over to our production team.

Kind regards,

on behalf of

Prof. Sabata Martino

Academic Editor

PLOS ONE